

# China Coastal GNSS Network: Advancing Precipitable Water Vapor Monitoring and Applications in Climate Analysis

Zhilu Wu[1], Bofeng Li[1], Qingyuan Liu[1], Yanxiong Liu[2], Huayi Zhang[2], Dongxu Zhou[2], Yang Liu[2]*

[1]College of Surveying and Geo-Informatics, Tongji University, 200092, Shanghai, China

5    [2]First Institute of Oceanography, Ministry of Natural Resources, 266061, Qingdao, China

*Correspondence to*: Yang Liu (yangliu@fio.org.cn)

**Abstract:** The Global Navigation Satellite System (GNSS) offers precise, continuous monitoring of atmospheric water vapor, essential for weather forecasting and climate research. This study presents a high-accuracy precipitable water vapor (PWV) dataset from 55 GNSS stations along China's coast (2009–2019). PWV retrievals utilized weighted mean temperature ($T_m$) and zenith hydrostatic delay (ZHD) derived from fifth-generation European ReAnalysis (ERA5) products. After rigorous quality control, the dataset achieved an average completeness rate of 70%. Validation against ERA5 PWV products showed strong agreement (mean bias: 0.80 mm; RMS error: 2.52 mm), while comparisons with radiosonde profiles yielded a mean bias of 0.90 mm and an RMS error of 3.01 mm, confirming its accuracy and reliability. Spatial analysis revealed PWV values ranging from 0 to 88.57 mm, with minima decreasing with increasing latitude and concentrated around the Yangtze River estuary. Temporal patterns exhibited prominent annual and semi-annual cycles, particularly in higher latitudes. PWV showed a strong positive correlation with sea surface temperature (SST; r = 0.76), with a 1 K SST increase leading to a 2.4 mm (7%) PWV rise. This dataset supports high-precision applications, including PWV validation, extreme weather prediction, and climate trend analysis. The processed ZTD and PWV datasets from 55 CGN stations are accessible at https://zenodo.org/records/14639032.

**Keywords:** Water Vapor, China Coastal, Ground-based GNSS networks, Spatiotemporal distribution, Sea surface temperature

## 1.Introduction

Water vapor is a crucial component of the atmosphere, playing a significant role in the energy budget, hydrological cycle, weather forecasting, and climate change (Galewsky et al., 2016; Dirmeyer and Brubaker, 2007; Karl and Trenberth, 2003). A precise understanding of atmospheric water vapor is essential for advancing knowledge in these areas. Water vapor can be measured using various techniques, including ground-based microwave radiometers (Elgered and Jarlemark, 1998), radiosondes (Ross and Elliott, 2001), and satellite remote sensing (Gao and Kaufman, 2003; Wu et al., 2024). The use of the Global Navigation Satellite System (GNSS) for water vapor sensing was first introduced by Bevis (Bevis et al., 1994; Bevis et al., 1992). Since then, numerous studies have been conducted, establishing GNSS as one of the primary methods for



measuring precipitable water vapor (PWV) (Wu et al., 2022b; Wang et al., 2016b).

Compared to other water vapor sensing techniques, GNSS meteorology (GNSS/MET) offers distinct advantages, including all-weather capability, low cost, and high temporal resolution. These features have made it a widely applied tool in numerical weather prediction assimilation (Gendt et al., 2004), extreme weather forecasting (Tsushima and Ohta, 2014). Additionally, due to its ability to provide continuous and high-accuracy PWV measurements, GNSS/MET has been extensively

utilized in climate change research based on long-term GNSS-derived PWV datasets. For instance, Wang et al. (2016a) analyzed global water vapor trends from 1988 to 2011 using GPS measurements. Alshawaf et al. (2017) evaluated climate evolution over Germany based on GNSS-derived PWV time series. Wang et al. (2018) investigated the relationship between GNSS-derived PWV and sea surface temperature (SST) as well as its response to El Niño–Southern Oscillation (ENSO).Yang et al. (2024) investigated the PWV spatiotemporal distribution across four major climate regions in China using GNSS data

from 248 stations over a 10-year period (2011–2020), which provided high temporal and spatial resolution for understanding regional atmospheric conditions. Their findings indicate that GNSS-derived PWV can serve as a valuable indicator for monitoring the evolution of climate change.

Therefore, many countries and organizations have established ground-based GNSS networks to monitor long-term variation in PWV. Examples include SuomiNet in the United States (Ware et al., 2000), European National Meteorological

Services (EUMETNET) E-GVAP system (Jones et al., 2023), the GEONET system operated by the Geospatial Information Authority of Japan (Sato et al., 2013) and ground-based observation network operated by China Meteorological Administration (CMA) (Bai et al., 2021). Recent researches utilized GNSS observations to generate high-accuracy PWV data, such as the EUREC4A project proposing water vapor products from 49 GNSS stations (Bosser et al., 2021), and Yuan et al. (2022) generated high-quality global PWV dataset from 12 552 ground-based GNSS stations in 2020.

Since 2009, the Ministry of Natural Resources (MNR) of China has launched and maintained the China coastal continuous operating GNSS Network (CGN). Initially designed to monitor land deformation at tidal stations and measure absolute sea level rise, the network also provides valuable data on water vapor exchange between oceans and land (Zhou et al., 2016; Zhou et al., 2022). Additionally, zenith total delays (ZTD) or PWV values derived from the CGN have been used to validate satellite remote sensing measurements (Zhu et al., 2021; Wu et al., 2022a; Chen et al., 2018), shipborne GNSS PWV estimates (Wu et

al., 2022b), typhoon event forecasting (Zhao et al., 2018), and long term climate analysis (Wang et al., 2017).

These studies underscore the diverse applications of atmospheric water vapor data derived from CGN observations. For instance, Wang et al. (2017) analyzed the spatiotemporal distribution of water vapor across 27 CGN stations using a six-year GNSS PWV dataset. However, comprehensive long-term evaluations of CGN PWV data across all stations remain limited, and the GNSS PWV retrieval methodologies require further refinement. To address these gaps, this study focuses on a decade-

long (2009-2019), high-resolution dataset specific to China's coastal regions, employing a refined PWV retrieval algorithm.

The processing methodology for China's coastal GNSS data is thoroughly detailed, and the dataset has undergone rigorous quality control to ensure reliability. This high-accuracy dataset offers significant potential for validating satellite remote sensing measurements, enhancing extreme weather prediction capabilities, and advancing long-term climate research.

This paper is structured as follows: Section 2 introduces the design and implementation of the CGN, along with the GNSS water vapor retrieval algorithm. Additionally, the fifth-generation European ReAnalysis (ERA5) data and radiosonde profiles used to evaluate the GNSS-derived PWV are described. Section 3 evaluates the performance of GNSS water vapor estimates and analyzes the spatial and temporal characteristics based on a decade-long GNSS PWV dataset. Finally, the summary and conclusions are presented in Section 4.

**2.Dataset and Methodology**

**2.1China Coastal Continuous Operating GNSS Network and Data Processing**

**2.1.1China Coastal Continuous Operating GNSS Network**

The CGN project was proposed in 2008 and completed in 2009 by the MNR of China. The mission of CGN includes monitoring global mean sea level, measuring water vapor, providing reference positions for coastal investigations, and offering early warnings for tsunamis and seabed earthquakes. To fulfill these responsibilities, the GNSS site locations were required to meet

the following criteria:

    (1). Proximity to tidal stations;

    (2). Open areas with sufficient satellite visibility at a cutoff angle of 5°;

    (3). Regions with stable geological structures;

    (4). Accessibility to power and communication networks.

The survey and site selection for the GNSS stations began in March 2008 and was completed in June 2008. Observations were collected using TPS NET-G3 GNSS receivers, with 24-hour data collection, and the TEQC software was used to assess data quality. Ultimately, 55 GNSS stations were established, and their geographical distributions are shown in Fig. 1 (blue circles in left panel). Fig. 2 illustrates the locations of the CGN stations, which are primarily situated close to the ocean. The stations cover regions with various climate types, including the temperate monsoon (in circles), subtropical monsoon (in

rectangles), and tropical monsoon climates (in triangles).

Most GNSS sites were initially equipped with Topcon TPS NET-G3 receivers featuring choke-ring antennas. At the outset, only GPS and GLONASS observations were available. Recent years, with the advancement of the Galileo and BeiDou-3 systems, as well as the upgrade of GNSS receivers at CGN sites, observations from four constellations have been collected. These multi-GNSS observations were archived and processed by GNSS Data and Analysis Center at the First Institute of



Oceanography (FIO), MNR of China.

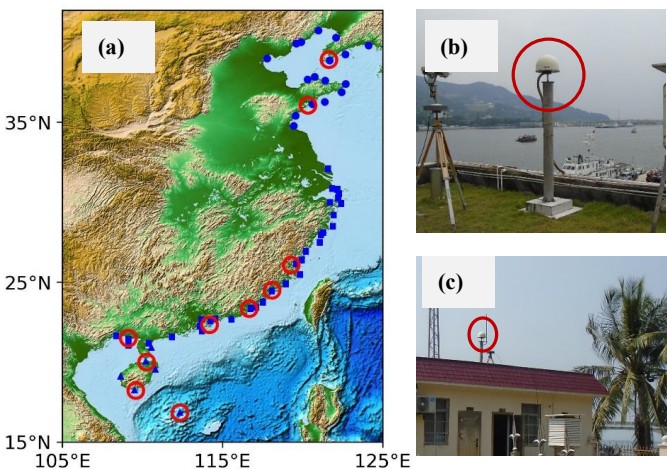

**Figure 1: Geographical distribution of the China Coastal GNSS network. (a) Overview of GNSS station locations. (b) and (c) Locations of two example CGN stations: NSZN and NXSA, respectively. In the left panel, blue circles indicate GNSS stations in temperate monsoon climates, blue rectangles represent stations in subtropical monsoon climates, blue triangles denote stations in**
**tropical monsoon climates, and red circles indicate radiosonde stations.**

**2.1.2 GNSS ZTDs Retrieval Algorithm**

GNSS observations with 30-s sampling rate from 55 GNSS stations were collected and processed by the FIO, MNR of China. FIO routinely processes GNSS observations by Positioning And Navigation Data Analyst (PANDA) released by Wuhan University (Shi et al., 2008; Liu and Ge, 2003) using static precise point positioning (PPP) model (Zumberge et al., 1997).

*a.*   **GNSS Data Processing Strategies**

The first-order ionospheric delays effect was removed by undifferenced ionospheric-free (IF) combinations of dual-frequency pseudorange and carrier phase observations. The cut-off angle of GNSS observation was 7°, and elevation-dependent weighting function was applied. The precise satellite orbits and clocks were from the European Space Agency (ESA) second reprocessing products (2009-2014) and operational final products (2014-2019). The errors caused by the solid Earth

tide, the ocean tide loading, and the pole tide were corrected based on the International Earth Rotation and Reference Systems Service (IERS) 2010 (Petit and Luzum, 2010), while the non-tidal ocean loading and atmosphere loading were not corrected. Antenna phase center offset (PCO) and phase center variation (PCV) were corrected by International GNSS service (IGS) antenna files(Schmid et al., 2007). The phase wind-up was also corrected according to Wu et al. (1993).

The tropospheric delays were modelled with a prior ZTD and global mapping function (GMF) (Böhm et al., 2006). The

ZTD consist of a hydrostatic part and a wet part, and the prior values were derived based on Saastamoinen model (Saastamoinen, 1972) with meteorological data (pressure and temperature) that provided by Global Pressure and Temperature



(GPT) (Böhm et al., 2007). The zenith wet delay (ZWD) was processed as an unknown piece-wise constant parameter with an interval of 60 mins. The tropospheric horizontal gradient was processed as an unknown piece-wise constant parameter with an interval of 720 mins (Chen and Herring, 1997). Batch least-squares estimator was used to estimating the GNSS station static

coordinate, epoch-wise clock offsets, and tropospheric delay.

### b. Outlier Detection

To ensure the reliability and accuracy of the retrieved GNSS ZTD data, a two-step screening process was conducted:

Step 1: General Range Check

A range-based screening was implemented to eliminate apparent outliers. Using hourly ERA5 ZTD values based on layer-

level pressure data for coastal areas in China from 2009 to 2019, the observed ZTD values ranged from 2278.3 mm to 2771.5 mm. As such, we set the ZTD range between 2000 mm and 3000 mm, resulting in the removal of 0.02% of GNSS ZTD values.

Step 2: Station-Specific Thresholds

Additionally, a moving window outlier detection approach, based on the interquartile range (IQR) rule proposed by (Yuan et al., 2022), was applied. The threshold for outlier detection was defined as Q1-3*IQR to Q3+3*IQR. The IQR represent Q3-

Q1, Q1 and Q3 are the 25th and 75th percentiles of all 5-minute ZTD data within a 15-day window centered on the target day. This moving window approach was implemented across all stations, leading to the removal of 0.29% of GNSS ZTD outliers.

### c. Validation of GNSS ZTDs based on ERA5 products

ERA5-derived ZTDs, known for their stability, were used to validate the GNSS ZTDs. Four GNSS stations (DBJO, DCHM, DDJS, and DPTN) exhibited significant abnormal ZTD differences in RMS when compared to the ERA5 ZTDs as

shown in Fig. 2. Although the average root mean square (RMS) difference between GNSS ZTD and ERA5 ZTD was less than 2 cm, the RMS errors for these four stations exceeded 3 cm, leading to the exclusion of GNSS ZTD data from these stations for the period under consideration.

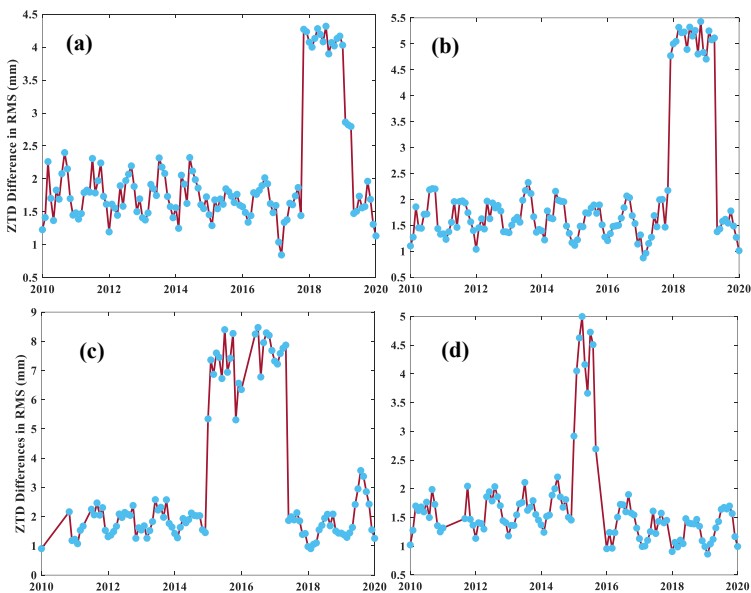

**Figure 2: Monthly differences in RMS values for DBJO (a), DCHM(b), DDJS(c), and DPTN(d)**

After applying the aforementioned screening criteria, we compared the GNSS ZTD estimates with the ERA5 ZTDs. As

illustrated in Fig. 3, the GNSS ZTDs demonstrate a strong agreement with the ERA5 ZTDs, exhibiting a root mean square

(RMS) difference of 1.52 cm and a mean bias of -0.04 cm. Notably, the concordance between GNSS ZTDs and ERA5 ZTDs

is significantly better in high-latitude regions, with the largest observed difference being 2.4 cm at the NXSA station, as

presented in Fig. 4, which depicts the differences across each station. Generally, GNSS ZTDs are lower than those from ERA5

at most GNSS stations, with the mean bias at the DDCN station registering at 1.7 cm.

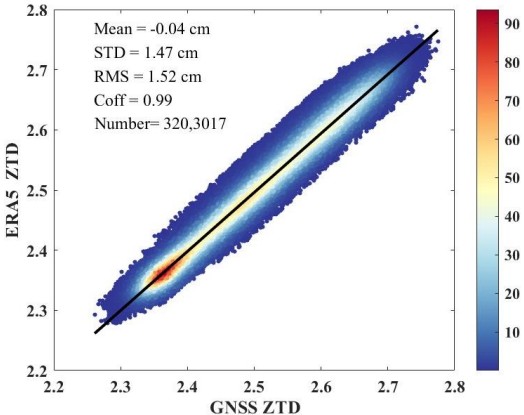

**Figure 3: Scatter Density of GNSS ZTDs and ERA5 ZTDs**
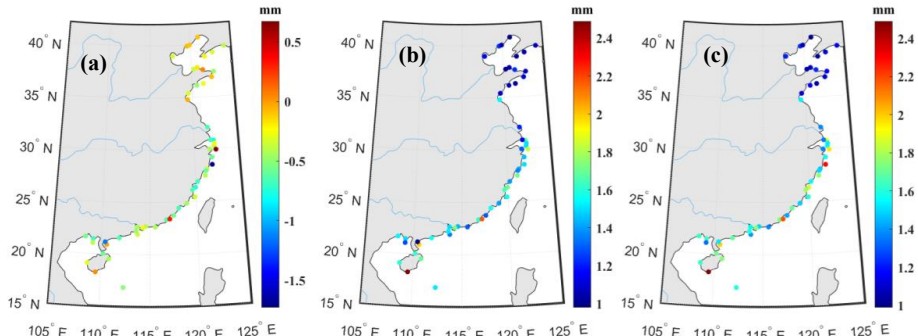

**Figure 4: Comparison between GNSS ZTDs and ERA5 ZTDs: (a) mean bias; (b) STD (c)RMS**

**2.1.3 Water Vapor Retrieval Algorithm**

After obtaining accurate GNSS ZTD estimates, PWV values are calculated based on the following equation:

$$PWV = Q \cdot (ZTD - ZHD) = Q \cdot ZWD \qquad (1)$$

$$Q = \frac{10^6}{(k_2' + \dfrac{k_3}{T_m}) \cdot R_v \cdot \rho_{lw}} \qquad (2)$$

where $\rho_{lw}$ denotes the liquid water density; $R_v$ represent the water vapor specific gas constant $R_v = (461.495\ J \bullet (kg \bullet K)^{-1})$; $k_2'$

$= (17 \pm 10)\ K \bullet hPa^{-1}$ and $k_3 = (3.776 \pm 0.004)10^5 K^2 \bullet hPa^{-1}$ are the atmospheric refractivity constants (Zhang et al., 2019). $T_m$ is

derived from the ERA5 layer data and is shown as follows:

$$T_m = \frac{\displaystyle\int_{H_0}^{\infty} \frac{e}{T} \cdot Z_w^{-1} \cdot dH}{\displaystyle\int_{H_0}^{\infty} \frac{e}{T^2} \cdot Z_w^{-1} \cdot dH} \qquad (3)$$

where $e$ (in $hPa$) is the water partial pressure, $Z_w$ is the wet term of refraction that taken as 1 (Schuler, 2001), and T (in $K$) is

the temperature in the layer.

The algorithm to derive ZHD values is based on ERA5 hourly data on 37 pressure levels with the following equation:

$$ZHD = 10^{-6} \sum_i (k_1 \times (P_i - e_i)/T)\Delta s_i \qquad (4)$$

$$e_i = h_i \times P_i / 0.622 \qquad (5)$$

where $k_1 = 77.604$ K/Pa, $P_i$ is the atmosphere pressure ($hpa$), $e_i$ is the vapor pressure ($hpa$), $T_i$ is the temperature ($K$), $h_i$

is the specific humidity.



### 2.1.4 Water Vapor Data Quality Control


A two-step screening process was also conducted to ensure the reliability and accuracy of the retrieved GNSS PWV data, which is similar to the ZTD screening method. A range-based screening was implemented to eliminate apparent outliers. Using 1-hour ERA5 PWV data from coastal areas of China during the period 2009–2019, the observed PWV values ranged between 0.72 mm and 86.21 mm. Based on this analysis, any PWV value below 0 mm or above 90 mm was considered an outlier and subsequently removed., which excluded 0.07% of the total dataset.


Besides, a station-specific outlier detection method was employed. This involved applying a moving IQR rule to detect anomalies. For each station, the median PWV value was calculated within a 15-day moving window centered on the specific day. The outlier thresholds were defined as: Q1-3*IQR to Q1+3*IQR. PWV values falling outside this range were flagged and removed. This step resulted in the rejection of an additional 0.26% of the data points across all stations.

### 2.1.5 China Coastal GNSS PWV Product


Fig. 5 summarizes the workflow for generating the GNSS-based PWV products for the China coastal region. The analysis utilized 30-second sampling rate GNSS observations from 55 coastal GNSS sites in China, covering the period from 2009 to 2019. Hourly GNSS ZTD estimates were obtained using the PANDA processing software. Following a screening process for the GNSS ZTD data, the ZHD and weighted-mean temperature were derived from ERA5 data, based on 37 pressure levels. This method has shown significantly better accuracy compared to the values derived from the Saastomoinen and Bevis equations (Yuan et al., 2022). The GNSS ZTD values were then converted into GNSS PWV estimates. Finally, the China coastal PWV products for the period 2009 to 2019 were generated, with further screening applied to ensure data quality.


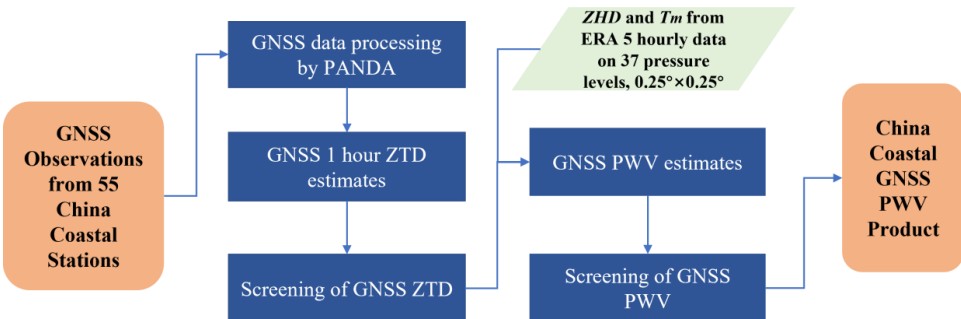

**Figure 5: Workflow of generating China Coastal GNSS PWV products**





### 2.2 Validation Dataset

#### 2.2.1 Numerical Weather Model

The numerical weather model used in this study is the ERA5 products, which is provided by the European Centre for Medium-Range Weather Forecasts (ECMWF). ERA5 is the fifth-generation reanalysis product, offering comprehensive data for the land, ocean, and atmosphere (Hersbach et al., 2020). In this study, a single-level product with a temporal resolution of 1 hour and a spatial resolution of $0.25° \times 0.25°$ was used to validate the GNSS PWV data. Additionally, ERA5 layer products with the same temporal and spatial resolutions were utilized to derive the GNSS ZHD and weighted-mean temperature. Bilinear interpolation was applied in the horizontal direction to extract the parameters corresponding to the coordinates of the GNSS stations. Furthermore, hourly SST data provided by ERA5 were used to investigate the relationship between SST and coastal PWV.

#### 2.2.2 Radiosonde Profile

In this study, radiosonde (RS) profiles were used to validate the GNSS PWV data. These profiles were obtained from the Integrated Global Radiosonde Archive (IGRA), which is accessible at (https://www.ncei.noaa.gov/products/weather-balloon/integrated-global-radiosonde-archive). IGRA is one of the most comprehensive radiosonde datasets, encompassing over 2800 stations worldwide. Of these, 770 stations provided regular daily observations at 00:00 and 12:00 UTC (Durre et al., 2018). A distance threshold of 30 km and a time threshold of 1 hour were applied to select the matching radiosonde stations. As a result, 10 radiosonde stations were chosen for validation, which are highlighted in red circles in Fig. 1. Detailed information about the radiosonde and GNSS stations is provided in Table 1.

Table 1 Detail Information of GNSS Stations and RS Stations

| RS Station | GNSS Station | Distance(km) | Start Time | End Time |
|---|---|---|---|---|
| 45004 | NSZN | 27.55 | 2009.12 | 2019.12 |
| 54662 | BLHT | 6.41 | 2009.12 | 2019.12 |
| 54857 | BXMD | 25.17 | 2009.12 | 2019.12 |
| 58847 | DCHM | 32.74 | 2009.12 | 2019.12 |
| 59134 | DXMN | 5.18 | 2011.11 | 2019.12 |
| 59316 | NSTO | 9.05 | 2009.12 | 2019.12 |
| 59644 | NBHI | 8.57 | 2014.05 | 2015.09 |
| 59758 | NHKO | 7.34 | 2009.12 | 2019.12 |
| 59948 | NSYA | 9.17 | 2014.05 | 2015.09 |
| 59981 | NXSA | 0.20 | 2012.06 | 2016.03 |

The temperature and relative humidity (RH) profiles were obtained from the IGRA dataset, and strict quality control criteria were applied to ensure data reliability. The following conditions were used for quality assurance (Zheng et al., 2022):

(1) The profile should extend from the surface to a pressure level of at least 300 $hPa$.



(2)  The profile should contain data for more than five pressure levels below 100 *hPa.*

(3)  The pressure gap between adjacent layers should not exceed 200 *hPa.*

(4)  Only profiles containing data at the mandatory levels (10 levels) and significant levels were considered.

After applying these quality control procedures to the radiosonde profiles, the water vapor was derived as follows:

$$PWV_{RS} = \int_{H_{GNSS}}^{H_{top}} q(P)/g(P)dp \tag{6}$$

**3.Results and Discussion**

**3.1 Evaluation of GNSS Water Vapor Estimates**

**3.1.1 Comparison with ERA5 PWV Products**

To validate the GNSS-derived PWV estimates, we compared them with single-level PWV products from ERA5. Prior to the comparison, a height correction was applied to the GNSS PWV data. The corresponding values for each GNSS station were obtained using bilinear interpolation in the horizontal dimension. Outliers in the GNSS PWV data were removed using the two-sigma rule. The results of this comparison are shown in Fig. 6. The GNSS PWV estimates exhibit excellent agreement

with the ERA5 PWV, with an RMS difference of 2.58 mm. On average, the GNSS PWV is slightly lower than the ERA5 PWV by 0.80 mm. Additionally, the correlation coefficient between the GNSS and ERA5 PWV is 0.99, indicating a very high level of consistency between the datasets.

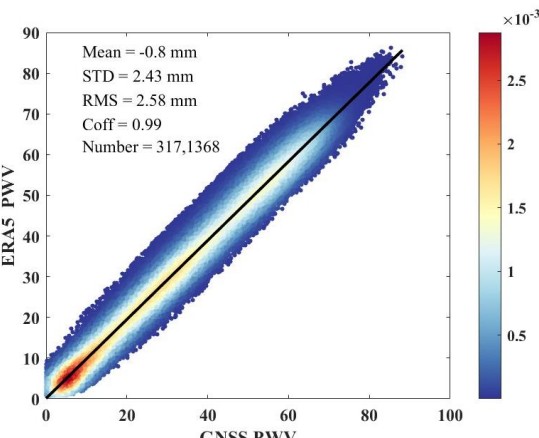

**Figure 6: Scatter Density of ERA5 PWV and GNSS PWV**

Additionally, the performance of GNSS-derived PWV at each site was evaluated, as shown in Fig. 7. The results indicate that GNSS sites located between 35° N and 45° N, covering the Bohai Sea and the North Huanghai Sea, exhibit significantly better agreement than other sites. In these regions, the mean biases are close to zero, and the standard deviation (STD) values are less than 2 mm. These areas, characterized by a temperate monsoon climate, show more stable variations in water vapor. Furthermore, the ionosphere in these regions is less active compared to other areas. In contrast, for GNSS sites located in



regions with subtropical and tropical monsoon climates, the discrepancies are more pronounced, with mean biases exceeding

1 mm and RMS differences greater than 2.5 mm.

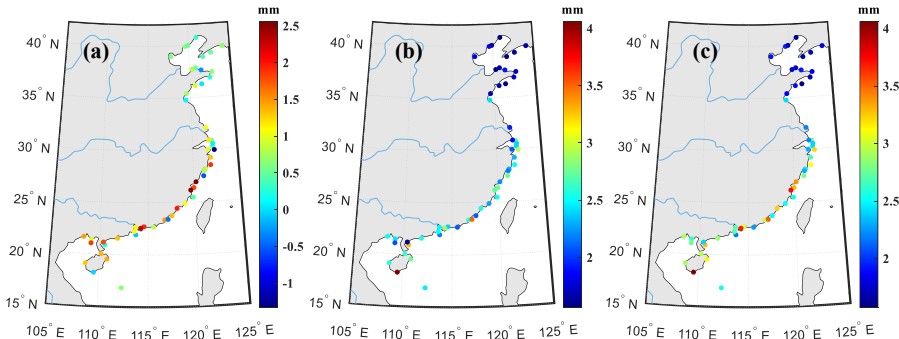

**Figure 7: Comparison between GNSS PWV and ERA5 PWV: (a) mean bias; (b) STD (c)RMS**

### 3.1.2 Comparison with Radiosonde Profiles

230    Furthermore, PWV data derived from RS profiles were used to evaluate the GNSS PWV estimates. As mentioned in Section

2.1.4, GNSS ZTDs from the DCHM dataset exhibit significant discrepancies compared to the ERA5 ZTDs. To assess the

accuracy of the GNSS PWV data, we compared the PWV estimates from the DCHM GNSS station with those from a nearby

RS station. The monthly differences in the RMS values are shown in Fig. 8. It is evident that during the period from October

2017 to March 2018, the monthly RMS differences exceeded 10 mm, which further suggests that the quality of the GNSS

observations from the DCHM station during this period is unreliable.

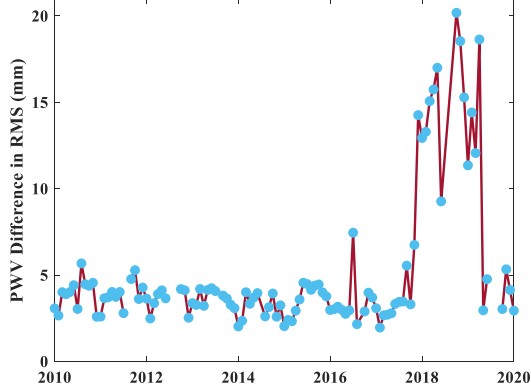

**Figure 8: Comparison between GNSS PWV and ERA5 PWV at DCHM**

The overall comparison between RS-derived PWV and GNSS PWV is presented in Fig. 9(a). The RMS difference is 3.49

mm, which is larger than the difference when compared to ERA5 PWV. This discrepancy may be attributed to the nature of

ERA5 data, which is a reanalysis product, while RS profiles provide only one or two observations per day. Additionally, the

quality of the RS profiles may contribute to the larger biases observed. As shown in Fig. 9(b), when the mandatory levels of

RS profiles are increased from 10 to 11, the RMS difference decreases to 3.01 mm. This suggests that stricter quality control





of RS profiles improves the agreement with GNSS PWV. However, this more stringent filtering significantly reduces the number of comparison points (see Table 2). Notably, RS profiles near the DXMN, NTSO, and NXSA stations show a marked

reduction in data points, indicating that the quality of the RS data from these stations is relatively poorer.

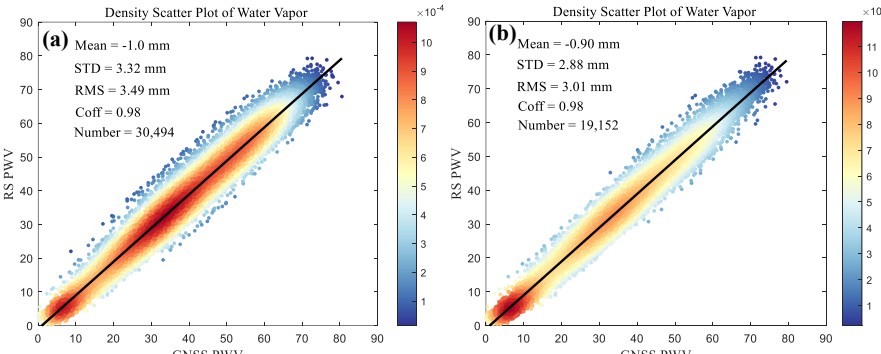

**Figure 9: Comparison between GNSS PWV and RS PWV with 10 levels (a) and 11 levels(b)**

Table 2 Comparison with GNSS PWV and RS PWV

| RS Station | GNSS Station | RS PWV with 10 levels | | | | RS PWV with 11 levels | | | |
|---|---|---|---|---|---|---|---|---|---|
| | | Mean (mm) | STD (mm) | RMS (mm) | Number | Mean (mm) | STD (mm) | RMS (mm) | Number |
| 45004 | NSZN | 0.48 | 2.08 | 2.14 | 4300 | 0.48 | 2.08 | 2.14 | 4300 |
| 54662 | BLHT | 1.38 | 2.06 | 2.48 | 3963 | 1.22 | 1.76 | 2.14 | 3209 |
| 54857 | BXMD | 1.44 | 2.13 | 2.57 | 3764 | 1.30 | 1.77 | 2.20 | 2575 |
| 58847 | DCHM | 1.80 | 3.40 | 3.85 | 4961 | 1.82 | 3.19 | 3.72 | 3439 |
| 59134 | DXMN | 1.67 | 2.66 | 3.14 | 1956 | 1.51 | 2.42 | 2.84 | 69 |
| 59316 | NSTO | 0.97 | 3.69 | 3.81 | 2915 | 0.29 | 2.60 | 2.56 | 24 |
| 59644 | NBHI | 0.00 | 3.07 | 3.07 | 1035 | -0.03 | 3.04 | 3.04 | 1020 |
| 59758 | NHKO | 1.39 | 3.81 | 4.06 | 4723 | 1.20 | 3.74 | 3.93 | 3702 |
| 59948 | NSYA | -0.45 | 4.73 | 4.75 | 821 | -0.46 | 4.73 | 4.75 | 814 |
| 59981 | NXSA | -0.64 | 5.45 | 5.48 | 2056 | / | / | / | / |

### 3.2 Spatiotemporal Characterizes of China Coastal GNSS PWV

After applying strict quality control to the GNSS PWV data from each station and validating the results using ERA5 products and RS profiles, we generated the China coastal GNSS PWV products. The spatial characteristics of the hourly GNSS PWV time series were analyzed using several metrics, including maximum value, minimum value, mean value, and the coefficient of variation (CV), defined as the ratio of the STD to the mean value. The completeness of the GNSS PWV data at each station is presented in Fig. 10. Most stations have a completeness rate exceeding 70%, with the average completeness of the GNSS

PWV products at 69.5%. However, the completeness for some stations is as low as 15%. To minimize the impact of missing data, only GNSS stations with data spanning at least three years and covering at least one full year were selected, resulting in a total of 51 coastal GNSS stations.

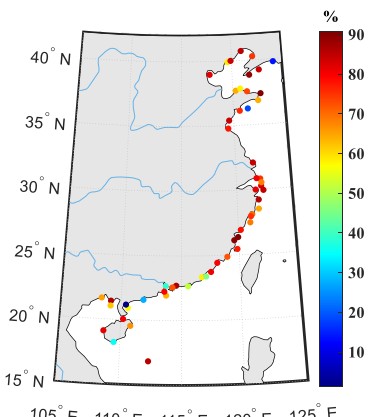

**Figure 10: Completeness of GNSS PWV at each station**

As shown in Fig. 11, the GNSS PWV time series range from 0 to 88.57 mm. The minimum values of China coastal GNSS

PWV generally decrease with latitude, with the lowest value of approximately 0.01 mm observed at the station near Dalian, as

shown in Fig. 10(b). The spatial distribution of the maximum values, however, is more complex. GNSS stations in the Bohai

Sea and North Yellow Sea exhibit the smallest maximum values, typically less than 80 mm. In contrast, the maximum values

of coastal GNSS PWV around the Yangtze River estuary are larger, with a peak of 88.57 mm at the DDJS station, located on

an island near Shanghai, during the super Typhoon Talim on September 15, 2017. This increase may be attributed to the higher

frequency of typhoons in this region. For other GNSS stations located between 30° N and 15° N, the maximum values range

from 80 mm to 84 mm.

The mean PWV values show a clear increasing trend as one moves toward the Equator. In the Bohai Sea, the mean values

are generally less than 20 mm, while in regions between 25° N and 15° N, the mean values are typically above 40 mm. The

lowest mean value, 15.3 mm, is observed at BQHD in Qinghuangdao, Hebei Province, while the highest mean value, 47.4 mm,

is found at NQLN in Qinglan, Hainan Province. In contrast, the CV values in the lower latitude regions are higher, with the

largest CV value of 90.7% observed at BZFD in Yantai. The lowest CV value, 27.5%, is found at NSZN in Shenzhen. This

suggests that the STD of PWV in areas with higher mean values is relatively smaller in proportion to the mean value.



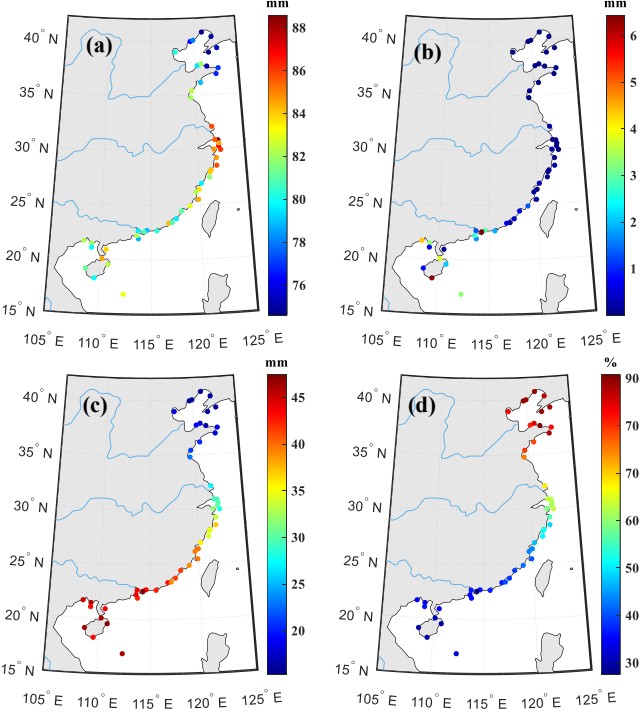

**Figure 11: Maxima (a), minima (b), mean values (c), and coefficients of variations (d) of hourly PWV products of China coastal GNSS stations.**

To analyze the temporal characteristics of GNSS PWV, six coastal GNSS stations were selected, ranging from high to low latitudes and representing three distinct climate patterns. Each station has a completeness rate exceeding 80%. Detailed information about the selected GNSS stations is provided in Table 3. The PWV time series and their fitted trends are shown in the left panel of Fig. 12, while the results of Fourier spectrum analysis are presented in the right panel. As shown in Fig. 12, the PWV time series at all stations exhibit a clear annual variation. However, stations located in high-latitude regions display a more pronounced semi-annual pattern in addition to the annual cycle. In contrast, the PWV time series at stations influenced by a tropical monsoon climate (Fig. 12e and 12f) predominantly show an annual cycle with no significant semi-annual component.








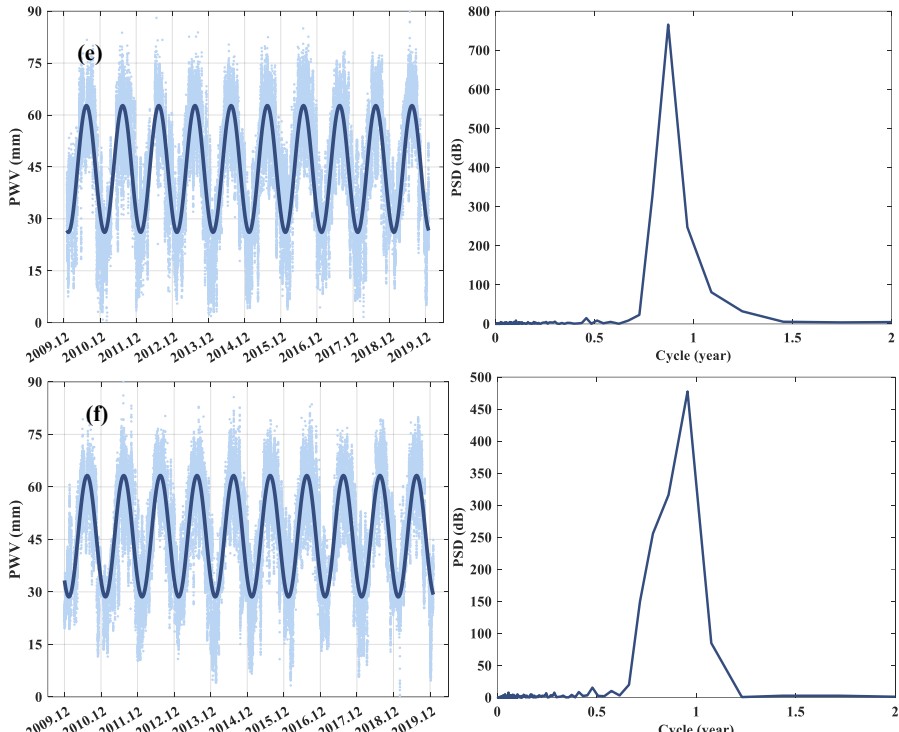

**Figure 12: PWV time series and corresponding spectrum analysis in BLHT(a), BTGU (b), DLSI (c), DSPU (d), NHZH (e), and NWZU (f).**


Table 3 Detail information of selected coastal GNSS stations

| Station Name | L | B | Climate Type | Completeness |
|---|---|---|---|---|
| BLHT | 38.9° N | 121.7° E | Temperate monsoon climate | 88.7% |
| BTGU | 38.9° N | 117.8° E | Temperate monsoon climate | 82.2% |
| DLSI | 32.1° N | 121.5° E | Subtropical monsoon climate | 84.3% |
| DSPU | 29.2° N | 121.9° E | Subtropical monsoon climate | 85.9% |
| NHZH | 22.7° N | 114.5° E | Tropical monsoon climate | 86.5% |
| NWZU | 16.8° N | 112.3° E | Tropical monsoon climate | 85.3% |

### 3.3 Relationship Between Sea Surface Temperature and coastal GNSS PWV

In the context of global warming, the climate impact of the ENSO has intensified. ENSO events are typically monitored using

SST, and variations in water vapor are strongly linked to ENSO activity. Therefore, we investigate the relationship between

coastal PWV from 51 GNSS stations with data spanning at least three years and nearby SST. The correlation coefficients

between GNSS PWV and SST are presented in Fig. 13. As observed, the correlation coefficients between PWV and SST

exceed 0.7 for all stations, with 7 stations showing values greater than 0.8. The mean correlation coefficient is 0.76, indicating

a strong positive relationship between the two datasets. Furthermore, the correlation coefficients for GNSS stations in

subtropical monsoon climates are generally smaller than those in areas with a temperate monsoon climate. This may be due to

the lower frequency of extreme weather events, such as typhoons, in the latter regions.

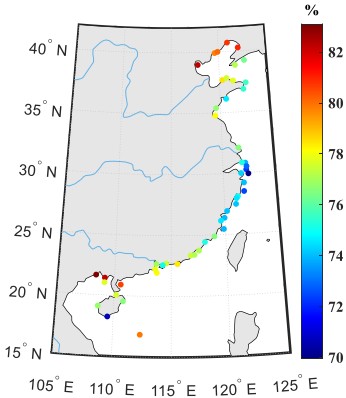

**Figure 13: Correlation coefficients between GNSS PWV and SST**

Additionally, we use six GNSS stations, as mentioned in Section 3.2, to illustrate the linear relationship between GNSS PWV and SST. The data clearly show a strong linear relationship between the two variables, with the slope of the relationship being steeper in lower-latitude areas. Furthermore, we calculated the increase in PWV associated with a 1 K rise in SST for

all coastal GNSS stations in China. As shown in Fig. 15(a), a 1 K increase in SST results in a PWV increase ranging from 1.5 mm to 4.5 mm, with an average increase of 2.4 mm. The PWV increase is more pronounced in low-latitude regions. Moreover, Fig. 15(b) shows that a 1 K SST increase causes an average 7% rise in PWV, with this percentage being relatively lower in areas with a subtropical monsoon climate.



Earth System
Science
Data

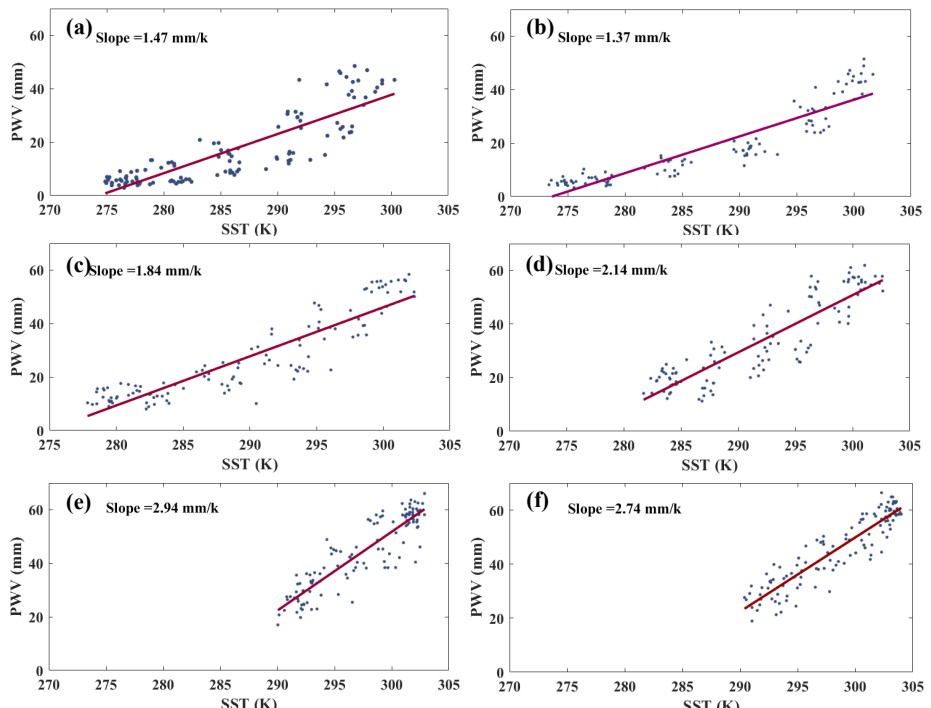

**Figure 14: Scatter plot for six GNSS stations: BLHT (a), BTGU (b), DLSI (c), DSPU (d), NHZH (e), NWZU (f).**

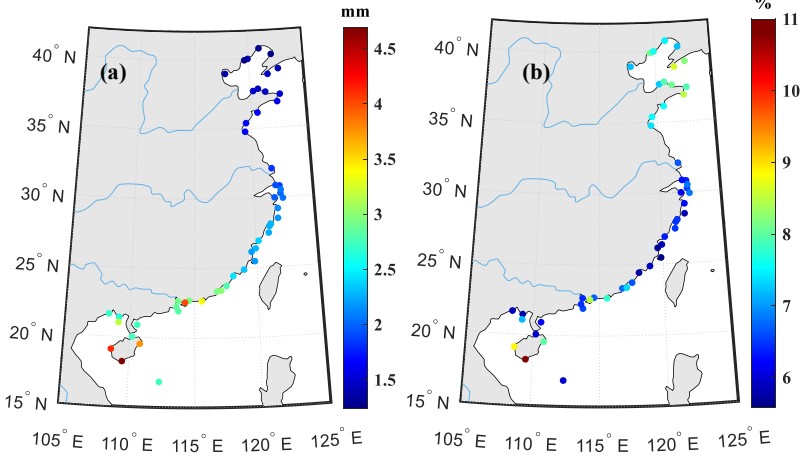

**Figure 15: PWV increasing value by 1K SST (a), and relative PWV increasing value by 1 K SST.**

## 4. Conclusions

GNSS has become an indispensable technique in numerical weather prediction assimilation, extreme weather forecasting, and climate change monitoring. Numerous countries and organizations have established GNSS-based ground networks for PWV monitoring. This study introduces the China coastal GNSS continuous observation system, established by the MNR. The strategies for processing GNSS data are outlined, and after rigorous quality control, the GNSS ZTDs demonstrate good





agreement with ERA5 products, exhibiting a mean bias of 0.04 cm and an RMS error of 1.5 cm. PWV estimates were derived based on $T_m$ and ZHD from hourly ERA5 pressure level data. A decade-long set of hourly GNSS PWV products (2009–2019) for 55 coastal GNSS stations was generated using a dedicated data screening algorithm. The GNSS PWV estimates exhibit a mean difference of 0.8 mm and an RMS error of 2.52 mm when compared to ERA5 data. A comparison with RS profiles, using at least 11 atmospheric levels, resulted in a mean bias of 0.9 mm and an RMS error of 3.0 mm. These results confirm the stability and high accuracy of the coastal GNSS PWV estimates.

Besides, the spatiotemporal characteristics of coastal PWV in China were analyzed. The PWV values ranged from 0 mm to 88.57 mm, observed during Typhoon Talim in 2017. The minimum values decrease with latitude, while the maximum values are concentrated around the Yangtze River estuary. The PWV time series exhibit a clear annual cycle, with a more pronounced semi-annual cycle in higher-latitude regions. Additionally, the relationship between coastal GNSS PWV and SST was analyzed, revealing a strong positive correlation, with a mean correlation coefficient of 0.76. Furthermore, a 1 K increase in SST leads

to a PWV increase ranging from 1.4 mm to 4.5 mm, with an average increase of 7% across the CGN stations.

Water vapor is a critical factor in weather forecasting and climate change analysis. This study highlights the significant potential of coastal GNSS PWV data for long-term climate pattern analysis, including ENSO, and its associated impacts.

**Data availability:** China coastal GNSS PWV products generated in this research is openly available at https://zenodo.org/records/14639032 (Wu and Li, 2025).

**Author contribution:** W.Z. and Y.L. designed the experiments. Q.L. developed the model code. Y.L., H. Z., and D. Z. validated the experiment results. W.Z. and B.L. prepared the manuscript with contributions from all co-authors.

**Competing interests:** The authors declare that they have no conflict of interest.

**Acknowledgment:** We thank the China coastal continuous operating GNSS Network of MNR, for GNSS observations, and GNSS Data and Analysis Center at First Institute of Oceanography, MNR for providing the GNSS ZTDs. We also thank ESA

for satellite precise clock and orbit products, ECMWF for ERA5 products, and IGRA for RS profiles. This research is financially supported by the National Key R&D Program of China (2023YFB3907201), Basic Scientific Fund for National Public Research Institutes of China (2022S03), and the National Natural Science Foundation of China (No. 42204018).

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
