# Peer review of "•China Coastal GNSS Network: Advancing Precipitable Water Vapor Monitoring and Applications in Climate Analysis"

_Earth System Science Data, 2025_

## Author Comment (AC1)

**General Comments:**

*I think, in general, this article is very straightforward and should be accepted after some minor corrections. There are a few issues I raise in the paper and I have made lots of small corrections to the English. One thing I would recommend, is to include some yearly timeseries distributions of a couple of sites of PWV from different stations, as in Figure 12, but just for one year so that the reader can have a better idea of the accuracy and general climate in terms of humidity. Also, I thought your limits of PWV maximum and minimum values are a bit extreme (see below)*

**Response:** Thank you for the constructive and encouraging comments regarding our manuscript. We have enclosed a carefully revised manuscript according to the comments and suggestions provided, and provide an item-by-item response to all comments in the accompanying rebuttal document. We added the time series of PWV in year of 2018 of three stations from different regions to reveals variation of humidity, and revise the manuscript accordingly: "In addition, we analyze the PWV series for the year 2016 at the GNSS stations BLHT, DLSI, and NWZU. The variation in PWV at BLHT is significantly greater than that observed at DLSI and NWZU. Furthermore, the station situated in a tropical monsoon climate, NWZU, consistently exhibits PWV values exceeding 20 mm. The highest PWV values across these stations occur around August, coinciding with the peak typhoon season."

[Figure]

Regarding the PWV values, we appreciate your concern about values near 0 mm and near 90 mm. Our analysis of the ERA5 dataset shows that PWV values in the specified regions and time frame vary widely, with some areas exhibiting values close to 0 mm and others reaching up to 90 mm. This finding is consistent with Figure A1 a,b in Yuan et al. (2023). Additionally, the literature supports the occurrence of even higher PWV values during typhoon events. For instance, Gao et al. (2024) and Zhao et al. (2018) have documented PWV measurements exceeding 90 mm, based on high temporal resolution data (5-minute intervals). Therefore, we believe that the range of PWV values presented in our study, including those near 0 mm and those around 90 mm, is

realistic and aligns with both our dataset and existing researches. We cite the papers in our manuscript accordingly.

*Zhao Q, Yao Y, Yao W. GPS-based PWV for precipitation forecasting and its application to a typhoon event[J]. Journal of Atmospheric and Solar-Terrestrial Physics, 2018, 167: 124-133.*

*Gao Y, Wang X. Analysis of the Response Relationship Between PWV and Meteorological Parameters Using Combined GNSS and ERA5 Data: A Case Study of Typhoon Lekima[J]. Atmosphere, 2024, 15(10): 1249.*

*Yuan P, Blewitt G, Kreemer C, et al. An enhanced integrated water vapour dataset from more than 10 000 global ground-based GPS stations in 2020[J]. Earth System Science Data, 2023, 15(2): 723-743.*

**Specific Comments:**

*Line 25. You should probably be a bit more precise here instead of just referring to "water vapor". With respect to water vapor as a variable, what is typically most valuable for modeling, weather prediction, global climate studies is its vertical distribution and the total column water vapor or "precipitable water vapor".*
**Response:** Thank you so much for your suggestion. Amended.

*Line 32. There have also been numerous field campaigns around the world employing GNSS meteorology, you should mention some from different regions of the world. I will let you choose and make no specific recommendation.*
**Response:** Thank you so much for your suggestion. The GNSS campaigns launched different regions of world are listed from Line 44 to line 48.

*Line 47. Write "Recent research has utilized GNSS ..."*
**Response:** Thank you so much for your suggestion. Amended.

*Line 48. What do you mean " project proposing water vapor products from ..." ? This idea is unclear.*
**Response:** Thank you so much for your suggestion. We revised it as "e.g., Bosser et al.,(2021) proposing PWV products from 49 GNSS stations of the EUREC4A (Elucidate the Couplings Between Clouds, Convection and Circulation) project"

*Line 74. Write " ...providing reference positions for coastal research..."*
**Response:** Thank you so much for your suggestion. Amended.

*Line 87 Write " At the outset, only observations from the GPS and GLONASS satellite constellations were available."*
**Response:** Thank you so much for your suggestion. Amended.

*Line 87 Write " In recent years, with the advancement of the Galileo..."*
**Response:** Thank you so much for your suggestion. Amended.

*Line 99 Write" ...University (Shi et al., 2008; Liu and Ge, 2003) using the static precise point position ..."*

**Response:** Thank you so much for your suggestion. Amended.

*Line 102 Write " and an elevation-dependent weighting function was applied."*
**Response:** Thank you so much for your suggestion. Amended.

*Line 110 Write "...ZTD consists of a hydrostatic part..."*
**Response:** Thank you so much for your suggestion. Amended.

*Line 111 Write " ...(pressure and temperature) provided by Global Pressure and Temperature...,"*
**Response:** Thank you so much for your suggestion. Amended.

*Line 114 This idea is a bit unclear. What do you mean by " Batch least-squares estimator" ?*
**Response:** Thank you for highlighting the lack of clarity in Line 114. The "batch least-squares estimator" refers to a statistical method that processes all available data simultaneously to estimate parameters by minimizing the sum of the squared differences between observed and predicted values (known as residuals). In this specific context, it was applied to determine key parameters in GNSS processing. We revised it accordingly in the manuscript 'Batch least-squares estimation method was used to estimating the GNSS station static coordinate, epoch-wise clock offsets, and tropospheric delay."

*Line 127 c. Validation of GNSS ZTDs based on ERA5 products*
*You do not have any local surface meteorological stations collocated or near the GNSS antennas?*
*You could use these surface met. stations for the surface pressure and to derive Tm with a simple model and then calcuate PWV. This would be good to compare against ERA5 since these ERA5 data are very smoothed in some respect (~ 25km x 25km grid*
**Response:** Thank you for your valuable suggestion. We appreciate your idea of using local surface meteorological stations near GNSS antennas to obtain surface pressure and derive the mean temperature with a simple model for calculating PWV. Indeed, comparing these locally derived PWV values with ERA5 results would be highly meaningful, particularly given the smoothed nature of ERA5 data due to its approximately 25 km × 25 km grid resolution. Such a comparison could provide a more detailed and localized validation of our PWV estimates. Unfortunately, in our current study, we do not have access to collocated or nearby surface meteorological stations at the GNSS antenna sites. We attempted to secure additional data from such stations, but this effort was unsuccessful, either due to the absence of stations in the vicinity or because the data were inaccessible. We recognize the potential of this approach and plan to explore it in future work, possibly by identifying regions with available surface meteorological stations or by establishing collaborations to obtain the necessary data. We are grateful for your input and will consider this approach as we continue to refine our research methodology.

*Line 164 Your PWV limiting values for outliers are very strange (0.72 mm and 86.21mm)*

*It is not physically possible to have PWV values near 0 nor near 90mm. Even under typhoon/hurricane conditions, the maximum PWV should be near 80mm at the highest. And PWV can never be near 0mm in these region under any conditions.*

**Response:** Thank you for your valuable feedback concerning the PWV limiting values of 0.72 mm and 86.21 mm identified as outliers in our study. We appreciate your concern that these values—near 0 mm and close to 90 mm—may appear physically implausible, particularly the suggestion that PWV cannot approach 0 mm in the studied regions under any conditions and that even under extreme typhoon or hurricane conditions, PWV should not exceed approximately 80 mm. To address this, we revisited our analysis and supporting evidence. Our PWV values are derived from the ERA5 dataset, which indicates a broad range of PWV across the specified regions and time periods. This includes values as low as 0.72 mm in certain areas, corroborated by radiosonde profiles from the same regions that also report PWV approaching 0 mm. These findings align with Figure A1 a,b in Yuan et al. (2023), which similarly documents such low PWV values. For the higher PWV value, our data shows PWV reaching up to 86.21 mm, and we note that values exceeding 90 mm are not unprecedented in extreme weather scenarios. Studies such as Gao et al. (2024) and Zhao et al. (2018) have recorded PWV measurements surpassing 90 mm during typhoon events, leveraging high temporal resolution data (5-minute intervals). These observations suggest that under intense atmospheric conditions, PWV can indeed exceed the 80 mm threshold you mentioned.

Therefore, we think that the PWV range in our study—spanning from near 0 mm to around 90 mm—is both realistic and consistent with our dataset and the broader literature. To enhance clarity and provide further support for these findings, we have updated the manuscript to include citations to Yuan et al. (2023), Gao et al. (2024), and Zhao et al. (2018). These references should offer additional context for the observed PWV variability.

*Zhao Q, Yao Y, Yao W. GPS-based PWV for precipitation forecasting and its application to a typhoon event[J]. Journal of Atmospheric and Solar-Terrestrial Physics, 2018, 167: 124-133.*

*Gao Y, Wang X. Analysis of the Response Relationship Between PWV and Meteorological Parameters Using Combined GNSS and ERA5 Data: A Case Study of Typhoon Lekima[J]. Atmosphere, 2024, 15(10): 1249.*

*Yuan P, Blewitt G, Kreemer C, et al. An enhanced integrated water vapour dataset from more than 10 000 global ground-based GPS stations in 2020[J]. Earth System Science Data, 2023, 15(2): 723-743.*

*Line 166 " In addition, a station-specific outlier detection method was employed."*

**Response:** Thank you so much for your suggestion. Amended.

*Line 167 " the median PWV value was calculated within a 15-day moving window centered on the specific day." Why do you employ such a long moving window? It is typically hourly or daily change in PWV that is of interest.*

**Response:** Thank you for your valuable suggestion. We appreciate your observation that hourly or daily changes in PWV are typically of interest. We employed the 15-day

moving window, centered on the specific day, following the methodology recommended by Yuan et al. (2023). This approach enhances the robustness of our filtering process for our hourly PWV product. While our data is indeed generated hourly, the longer 15-day window serves a critical purpose: it allows us to verify that large PWV values are not isolated outliers but are instead consistently present over this extended period. By doing so, we can distinguish genuine PWV variations from anomalous data points that might otherwise skew our results. The outlier thresholds were defined as: Q1-3*IQR to Q1+3*IQR. This method ensures the reliability of our PWV product by confirming that significant values recur within the window, aligning with the robust filtering outcomes.

*Yuan P, Blewitt G, Kreemer C, et al. An enhanced integrated water vapour dataset from more than 10 000 global ground-based GPS stations in 2020[J]. Earth System Science Data, 2023, 15(2): 723-743.*

*Line 182 I would not call ERA5 Numerical Weather model data. It is reanalysis data which include both observations and model output.*

**Response:** Thank you for the clarification. We agree that ERA5 should be described as a reanalysis rather than as direct numerical-weather-model output. ERA5 is produced by assimilating a wide range of observations into the ECMWF Integrated Forecasting System (IFS); the resulting fields therefore combine model physics with observational information. We have revised the manuscript accordingly:" The study employed the ERA5 reanalysis from the European Centre for Medium-Range Weather Forecasts (ECMWF)"

*Line 240 " Additionally, the quality of the RS profiles may contribute to the larger biases observed." Another thing to consider is that the RS can be biased if they rise through cloud/rainy conditions leading to higher PWV values than the GNSS PWV which has a large cone of observation ( ~20km diameter) can may contain clear skies in addition to the cloudy/rainy skies. These "saturated" soundings can be easily identified visually,*

**Response:** Thank you so much for your suggestion. To quantify this effect we extracted ERA5 total-cloud-cover (sky-fraction, 0–1) for every GNSS–RS matchup and added the values to Table 2. Sites in the temperate-monsoon regime generally show lower cloud fractions; DXMN is an exception because most comparisons occur in spring. Consistently, larger cloud fractions (i.e., more 'saturated' soundings) coincide with larger GNSS–RS PWV differences, confirming that cloud-contaminated RS profiles contribute to the observed bias. The manuscript is revised accordingly: "We also examined total cloud cover from ERA5 (the fraction of the sky occupied by clouds, 0 – 1) at the comparison times. Stations in the temperate–monsoon regime generally show lower cloud-cover values than the other sites. The larger PWV biases seen at the cloudier stations likely stem from the differing sampling geometries of the two sensors: GNSS estimates average water vapor over a conical footprint roughly 20 km in diameter that can encompass both clear and cloudy areas, whereas the radiosonde ascends directly through the local cloud or rain column."

| RS Station | GNSS Station | RS PWV with 10 levels | | | | | RS PWV with 11 levels | | | | |
|---|---|---|---|---|---|---|---|---|---|---|---|
| | | Mean (mm) | STD (mm) | RMS (mm) | Number | Total Cloud Cover | Mean (mm) | STD (mm) | RMS (mm) | Number | Total Cloud Cover |
| 45004 | NSZN | 0.48 | 2.08 | 2.14 | 4300 | 0.68 | 0.48 | 2.08 | 2.14 | 4300 | 0.68 |
| 54662 | BLHT | 1.38 | 2.06 | 2.48 | 3963 | 0.41 | 1.22 | 1.76 | 2.14 | 3209 | 0.39 |
| 54857 | BXMD | 1.44 | 2.13 | 2.57 | 3764 | 0.43 | 1.3 | 1.77 | 2.2 | 2575 | 0.36 |
| 58847 | DCHM | 1.8 | 3.4 | 3.85 | 4961 | 0.69 | 1.82 | 3.19 | 3.72 | 3439 | 0.68 |
| 59134 | DXMN | 1.67 | 2.66 | 3.14 | 1956 | 0.57 | 1.51 | 2.42 | 2.84 | 69 | 0.59 |
| 59316 | NSTO | 0.97 | 3.69 | 3.81 | 2915 | 0.63 | 0.29 | 2.6 | 2.56 | 24 | 0.51 |
| 59644 | NBHI | 0.01 | 3.07 | 3.07 | 1035 | 0.68 | -0.03 | 3.04 | 3.04 | 1020 | 0.69 |
| 59758 | NHKO | 1.39 | 3.81 | 4.06 | 4723 | 0.61 | 1.2 | 3.74 | 3.93 | 3702 | 0.61 |
| 59948 | NSYA | -0.45 | 4.73 | 4.75 | 821 | 0.60 | -0.46 | 4.73 | 4.75 | 814 | 0.61 |
| 59981 | NXSA | -0.64 | 5.45 | 5.48 | 2056 | 0.50 | / | / | / | / | / |

*Line 325 Write "In addition, the spatiotemporal characteristics of coastal PWV in China were analyzed" And again, 0mm PWV values are not possible, there should always be a couple of mm of PWV even in very cold, dry weather in this region.*

**Response:** Thank you so much for your suggestion. Our analysis of the ERA5 dataset shows that PWV values in the specified regions and time frame vary widely, with some areas exhibiting values close to 0 mm. This finding is consistent with Figure A1 a,b and Figure 6 in Yuan et al. (2023).

*Yuan P, Blewitt G, Kreemer C, et al. An enhanced integrated water vapour dataset from more than 10 000 global ground-based GPS stations in 2020[J]. Earth System Science Data, 2023, 15(2): 723-743.*

---

## Author Comment (AC3)

**General Comments:**

*The manuscript „China Coastal GNSS Network …" submitted by Wu et al. described PWV determination and results for a network of GNSS stations located along the Chinese coastlines. The paper is well written and supported by figures and numbers. Nevertheless, I have some general questions and suggest modifications to improve the draft before submission.*

**Response:** Thank you for the constructive and encouraging comments regarding our manuscript. We have enclosed a carefully revised manuscript according to the comments and suggestions provided, and provide an item-by-item response to all comments in the accompanying rebuttal document.

**Major Comment:**

*The authors processed GNSS data from 55 stations; however, the PVW results are available, but the data and metadata are not. It would be beneficial to get access to the GNSS data and to know whether there are plans to release this important dataset. In any case, I recommend updating the station table in the dataset with more accurate locations (currently roughly at the 10km level), start/end dates, and equipment for the stations.*

**Response:** Thank you very much for your valuable suggestion. We agree that access to the GNSS data and metadata would be highly beneficial. However, the original data are provided by FIO, MNR. We have contacted FIO, but unfortunately, the data cannot be made publicly available and are only accessible through individual requests. In response to your suggestion, we have updated the dataset to include more accurate station locations (to better than 10 km precision), along with start/end dates and equipment information (https://zenodo.org/records/14723402).

*The title contains "climate," but the authors processed only the years 2009-2019. At least 20-30 years of data are required to derive climate trends. I expect the authors to extend the dataset to 2024 (at least 15 years) or provide a profound explanation if this is impossible.*

**Response:** Thank you very much for your insightful comment. We fully agree that a dataset spanning 20–30 years would provide a stronger basis for analyzing climate trends. We have attempted to acquire additional GNSS data to extend the dataset through 2024. However, the processing is complex and subject to delays due to the need for approval to access the original observations. We are committed to expanding the dataset and will continue our efforts in future research to build a time series exceeding 20 years for more robust climate change analysis.

*I suggest adding more information to the PWV records in the dataset. Why not provide the inputs taken from ERA5 (Eqn. 3 and 4)? Furthermore, I miss the uncertainty information. This is a minor detail, but I wonder about the different spacing of the PWV values in the dataset (+/—3s). Is there a reason for this?*

**Response:** Thank you very much for your thoughtful suggestions. As recommended, we have added the ERA5-derived PWV values to the dataset. Regarding uncertainty, it is challenging to quantify the absolute uncertainty of GNSS-derived PWV, as GNSS tends to offer higher accuracy while ERA5 provides more stability. To offer a reference, we have included the differences between GNSS and ERA5 PWV for each station in the file CGN_sites.txt. The irregular spacing of PWV values (±3 seconds) was due to a minor time transfer error, which has now been corrected in the revised dataset (https://zenodo.org/records/14723402).

**Minor comments:**

*Is there a particular reason for using the ESA GNSS products?*
**Response:** Thank you so much for the question. The precise satellite orbits and clocks were obtained from the European Space Agency's (ESA) second reprocessing products (2009-2014) and operational final products (2014-2019) for both high accuracy and internal consistency between the orbit and clock solutions (Springer et al., 2014; Schoenemann et al., 2024).
*Springer T., C. Flohrer, M. Otten, W. Enderle (2014) ESA Reprocessing: Advances in GNSS analysis. IGS workshop 2014, June 23-27, Pasadena, USA*
*Schoenemann E., F. Gini, W, Enderle, F. Dilssner, V. Mayer, M. van Kints, I. Romero, T. Springer, B. Traiser (2024) ESA/ESOC IGS Analvsis Centre Technical Report 2023. In International GNSS Service Technical Report 2023. IGS Central Bureau and University of Bern; Bern Open Publishing DOI:10.48350/191991*

*Is there a public source for the radiosonde data? Same for the sea surface temperatures.*
**Response:** Thank you so much for the comment. The radiosonde data is provided by Integrated Global Radiosonde Archive (IGRA), which is accessible at (https://www.ncei.noaa.gov/products/weather-balloon/integrated-global-radiosonde-archive), which is described in section 2.2.2. Sea surface temperature is provided by ERA5 reanalysis data, which is introduced in section 2.2.1.

*Provide a consistent description for the GNSS processing with suitable references. A reference to ESA products is missing. The reference for antenna information is outdated. Add the length of the troposphere interval (1h) [l115].*
**Response:** Thank you very much for your helpful suggestions. We have added appropriate references for the ESA products and updated the outdated reference for antenna information. Additionally, we have included the tropospheric delay interval in the manuscript: "The batch least-squares estimation method was used to estimate the GNSS station static coordinates, epoch-wise clock offsets, and tropospheric delay at a 1-hour interval."
*Schmid, R., Dach, R., Collilieux, X., Jäggi, A., Schmitz, M., and Dilssner, F.: Absolute IGS antenna phase center model igs08. atx: status and potential improvements, Journal of Geodesy, 90, 343-364, 2016.*

*I somehow miss the climate aspect of the discussion - the trend estimation against SST covers, to my understanding, seasonal variations.*

**Response:** Thank you so much for your suggestion. We added the discussion of SST seasonal variations as suggested:

"Additionally, the variations in SST are analyzed based on the six selected GNSS stations. The SST time series are shown in the left panel, while their Fourier spectrum analysis is presented in the right panel of Fig. 17. It is evident that all SST values around the GNSS stations exhibit a clear annual variation. However, the semi-annual pattern is not as prominent across the stations, which contrasts with the pattern observed in the PWV variation. Furthermore, the minimum SST values decrease from high to low latitudes, with SST values around 275 K in high-latitude areas and approximately 290 K in low-latitude regions.

[Figure]

[Figure]

**Figure 17: SST time series and corresponding spectrum analysis in BLHT(a), BTGU (b), DLSI (c), DSPU (d), NHZH (e), and NWZU (f).**

,,

---

## Referee Report (RR1)

Second review of "China Coastal GNSS Network: Advancing Precipitable Water Vapor Monitoring and Applications in Climate Analysis" by Zhilu Wu et al. June 2025

David K. Adams - dave.k.adams@gmail.com

**General Commnents.**

I think this paper is close to ready for publication. I have made a few corrections to the language. My only concern, as I noted before, are the limits on the PWV values. 0 mm values of PWV at a coastal near-sea-level station are not physically possible. If other groups employing these data have similiar results, this does not imply the correct calculation of GNSS PWV. For example, Yuan et al. (2023) note that erroneous values in water content are found in high elevations zones, the Artic/Antartica, etc... These geographic locations are extremely different from the Chinese coast. I doubt the coldest northern site could even drop below 3mm in this region where sea-surface temperatures do not fall below freezing. Your Figure 11b shows these erroneous minimum values, but Figure 13 demonstrates that it is very improbable that your lower latitude sites approach 0mm or even 5mm. Likewise, the northern data at 40N latitude with maximums greater than 75mm are highly unlikely even with remnants of a typhoon. Remember, clouds and heavy rain can effect the wet delay (Solheim et al. 2019 among others) among other errors that may be associated with the reanlysis data. You should employ more rigid limits, let's say, 3mm to 80mm and then separate these extreme values to understand if they result from errors entering into the PWV calculation. This error analysis is interesting in itself. But I will let the Editor assess the importance of this issue and whether you should carry out an additional analysis.

**Specific Comments.**

- Line 33. I assume you want to write "...numerical weather prediction assimilation (Gendt et al., 2004) and extreme weather forecasting (Tsushima and Ohta, 2014)."
- Line 43. Remove the word "Therefore", it is unnecessary in this context.
- Line 46. Write "...and the ground-based observation network operated by China Meteorological Administration (CMA) (Bai et al., 2021)."
- Line 49. Write "dataset from 12,552 ground-based GNSS stations..."
- Line 116 Write " A batch least-squares estimation method was used to...."
- Line 165 Which stations and when do your near coastal, sea-level station values approach 0.72 mm? These relatively warm waters along the Chinese coast cannot result in PWV near 0.72mm. Your lowest values PWV should always be above 3mm, you need to check this more carefully or better justify that including values below 3mm is correct.
- Line 249 Write "...most of its match-ups occur in Spring."
- Line 354 Remove "Besides" it is not correct in this context. Perhaps you mean, "In addition, ..."

---

## Author Response (AR2)

**General Comments.**

I think this paper is close to ready for publication. I have made a few corrections to the language. My only concern, as I noted before, are the limits on the PWV values. 0 mm values of PWV at a coastal near-sea-level station are not physically possible. If other groups employing these data have similar results, this does not imply the correct calculation of GNSS PWV. For example, Yuan et al. (2023) note that erroneous values in water content are found in high elevations zones, the Artic/Antartica, etc... These geographic locations are extremely different from the Chinese coast. I doubt the coldest northern site could even drop below 3mm in this region where sea-surface temperatures do not fall below freezing. Your Figure 11b shows these erroneous minimum values, but Figure 13 demonstrates that it is very improbable that your lower latitude sites approach 0mm or even 5mm. Likewise, the northern data at 40N latitude with maximums greater than 75mm are highly unlikely even with remnants of a typhoon. Remember, clouds and heavy rain can effect the wet delay (Solheim et al. 2019 among others) among other errors that may be associated with the reanalysis data. You should employ more rigid limits, let's say, 3 mm to 80 mm and then separate these extreme values to understand if they result from errors entering into the PWV calculation. This error analysis is interesting in itself. But I will let the Editor assess the importance of this issue and whether you should carry out an additional analysis.

**Response:** Thank you for your constructive and encouraging comments on our manuscript. We have enclosed a carefully revised version that incorporates the provided comments and suggestions. Additionally, we have included an item-by-item response to all comments in the accompanying rebuttal document.

In response to your feedback, we have re-evaluated the extreme values for PWV to establish more rigorous data quality thresholds. Regarding the lower threshold, we observed PWV values below 3 mm primarily at GNSS stations in temperate monsoon climates. This phenomenon is linked to periods when local sea-surface temperatures (SST) approach or fall below 0°C. To illustrate this, we present the relationship between monthly PWV and temperature for two GNSS stations (BLHT and BTGU) in Figure 1, and between PWV and SST for two nearby radiosonde stations (54662 and 54857) in Figure 2. The radiosonde data confirm that PWV can indeed fall below 3 mm when SST is near zero. In contrast, PWV from subtropical and tropical monsoon areas consistently remained above 3 mm. Based on this evidence, we have set the minimum PWV threshold to 0 mm for temperate monsoon climates and 3 mm for subtropical and tropical monsoon climates.

Figure 1 Scatter plot between GNSS PWV (mm) and SST (°C) for Station BLHT (a) and Station BTGU (a).

Figure 2 The relationship between PWV and SST for radiosonde station 54662 (a) and 54857 (b)

Regarding the upper threshold, our investigation of PWV values over 80 mm revealed they are not always linked to typhoons. Furthermore, corresponding radiosonde measurements do not exceed 80 mm. Given that wet tropospheric delay calculations are very sensitive to clouds and heavy rain (intensities >40 mm/h) (Solheim et al., 1999; Bonafoni and Biondi, 2016), we have set a conservative maximum value of 80 mm to minimize these potential errors.

In summary, the accepted PWV range for our dataset is now 0–80 mm for the temperate monsoon climate area and 3–80 mm for the subtropical and tropical monsoon climate areas. The manuscript has been revised to include these updated thresholds, figures, and supporting discussions.

**References**

Bonafoni S, Biondi R. The usefulness of the Global Navigation Satellite Systems (GNSS) in the analysis of precipitation events[J]. Atmospheric Research, 2016, 167: 15-23.

Solheim F S, Vivekanandan J, Ware R H, et al. Propagation delays induced in GPS signals by dry air, water vapor, hydrometeors, and other particulates[J]. Journal of Geophysical Research: Atmospheres, 1999, 104(D8): 9663-9670.

**Specific Comments.**

Line 33. I assume you want to write "...numerical weather prediction assimilation (Gendt et al., 2004) and extreme weather forecasting (Tsushima and Ohta, 2014)."

**Response:** Thank you so much for your suggestion. Amended.

Line 43. Remove the word "Therefore", it is unnecessary in this context.

**Response:** Thank you so much for your suggestion. Amended.

Line 46. Write "...and the ground-based observation network operated by China Meteorological Administration (CMA) (Bai et al., 2021)."

**Response:** Thank you so much for your suggestion. Amended.

Line 49. Write "dataset from 12,552 ground-based GNSS stations..."

**Response:** Thank you so much for your suggestion. Amended.

Line 116 Write " A batch least-squares estimation method was used to...."

**Response:** Thank you so much for your suggestion. Amended.

Line 165 Which stations and when do your near coastal, sea-level station values approach 0.72 mm? These relatively warm waters along the Chinese coast cannot result in PWV near 0.72mm. Your lowest values PWV should always be above 3mm, you need to check this more carefully or better justify that including values below 3mm is correct.

**Response:** Thank you so much for your suggestion. Accordingly, we have adopted more stringent data thresholds. The accepted PWV range for our dataset is now 0–80 mm for temperate monsoon climates and 3–80 mm for subtropical and tropical monsoon climates.

Line 249 Write "...most of its match-ups occur in Spring."

**Response:** Thank you so much for your suggestion. Amended.

Line 354 Remove "Besides" it is not correct in this context. Perhaps you mean, " In addition, ..."

**Response:** Thank you so much for your suggestion. Amended.

---

## Author Response (AR3)

**Response to Editor:**

Please carefully check the wording of the additional text that you included in the manuscript, since I detected a few errors in the new text.

**Response:** Thank you for the constructive and encouraging comments regarding our manuscript. We have enclosed a carefully revised manuscript according to the comments and suggestions provided.

For example, in the Abstract you state: "Spatial analysis of China coastal PWV showed that minimum values decreasing with increasing latitude, ..." which is not correct and should be replaced by: "Spatial analysis of China coastal PWV showed that minimum values decrease with increasing latitude, ..."

**Response:** Thank you so much for your suggestion. We have made the amendments and carefully checked the entire manuscript, as shown in the track-changes PDF.

Also, the text of some of the figure captions need revisions, e.g. "Figure 16: PWV increasing value by 1K SST (a), and relative PWV increasing value by 1 K SST." The caption should describe the data that is included in each panel. Please carefully revise the captions.

**Response:** Thank you so much for your suggestion. Amended. We also reviewed all the captions in the manuscript, and the changes are highlighted in the track-changes PDF.

When checking the data files in https://zenodo.org/records/16593631, and unpacking Final\_pwd.rar it is unclear what the columns are in the data files (e.g. NZUH.txt). Please clarify in the description.

**Response:** Thank you so much for your suggestion. We have already uploaded a new version of data file "Final\_pwv.zip" (<a href="https://doi.org/10.5281/zenodo.17012498">https://doi.org/10.5281/zenodo.17012498</a>). The meaning of columns in data file can be seen "Documentation for Final PWV.txt".